# A novel machine learning approach to predict the export price of seafood products based on competitive information: The case of the export of Vietnamese shrimp to the US market

Nguyen Minh Khiem[1,2]*, Yuki Takahashi[3], Hiroki Yasuma[3], Khuu Thi Phuong Dong[4], Tran Ngoc Hai[5], Nobuo Kimura[3]

1 Graduate School of Fisheries Sciences, Hokkaido University, Hakodate, Hokkaido, Japan, 2 College of Information and Communication Technology, Can Tho University, Can Tho, Vietnam, 3 Faculty of Fisheries Sciences, Hokkaido University, Hakodate, Hokkaido, Japan, 4 School of Economics, Can Tho University, Can Tho, Vietnam, 5 College of Aquaculture and Fisheries, Can Tho University, Can Tho, Vietnam

* nmkhiem@cit.ctu.edu.vn

## Abstract

Predicting the export price of shrimp is important for Vietnam's fisheries. It not only promotes product quality but also helps policy makers determine strategies to develop the national shrimp industry. Competition in global markets is considered to be an important factor, one that significantly influences price. In this study, we predicted trends in the export price of Vietnamese shrimp based on competitive information from six leading exporters (China, India, Indonesia, Thailand, Ecuador, and Chile) who, alongside Vietnam, also export shrimp to the US. The prediction was based on a dataset collected from the US Department of Agriculture (USDA), the Food and Agriculture Organization of the United Nations (FAO), and the World Trade Organization (WTO) (May-1995 to May-2019) that included price, required farming certificates, and disease outbreak data. A super learner technique, which combined 10 single algorithms, was used to make predictions in selected base periods (3, 6, 9, and 12 months). It was found that the super learner obtained results in all base periods that were more accurate and stable than any candidate algorithms. The impacts of variables in the predictive model were interpreted by a SHapley Additive exPlanations (SHAP) analysis to determine their influence on the price of Vietnamese exports. The price of Indian, Thai, and Chinese exports highlighted the advantages of being a World Trade Organization member and the disadvantages of the prevalence of shrimp disease in Vietnam, which has had a significant impact on the Vietnamese shrimp export price.

**Data Availability Statement:** All relevant data are within the paper and its Supporting Information files.

**Funding:** This work was supported by the Hokkaido University DX Doctoral Fellowship [grant number JPMJSP2119]. The funders had no role in study design, data collection and analysis, decision to publish, or preparation of the manuscript.

**Competing interests:** The authors have declared that no competing interests exist

# Introduction

Shrimp production and export is an important economic activity in Vietnam. About 90% of Vietnamese shrimp is used for export, with a value of 2 billion USD in 2011 [1]. According to the Food and Agriculture Organization of the United Nations (FAO) [2], Vietnam is the world's third largest producer of farmed white leg shrimp and giant tiger prawn, accounting for 13% of the global total, while China is the largest (32%) and Indonesia is second (15%), followed by India (12%), Ecuador (9%), and Thailand (6%). Exports of seafood are important not only for economic growth but also for rural development and the improvement of livelihoods [3]. The EU, USA, and Japan are the three main importers of Vietnamese shrimp product, together accounting for more than 50% of the total export value. These markets have strict requirements for imported seafood products based on safe food criteria, traceability, and quality assurance.

Vietnam competes with other exporters to access the global market. Thus, Vietnam tries to satisfy mandatory requirements, seize other competitive advantages, and minimize shrimp disease. Food safety measures imposed by developed countries directly affect the trade flows of export countries [4]. Food safety requirements are met by global GAP certification and the Safe Quality Food standard [5]. Traceability based on the Hazard Analysis Critical Control Point is required in the agriculture chain to increase the value of the product [6]. Country of Origin Labeling and Aquaculture Steward Council certificates are also used to indicate the quality of exported products. Since 2003, the US market has imposed an anti-dumping measure on shrimp imports from China, India, Thailand, Vietnam, Ecuador, and Brazil, which is believed to have affected the export price of those countries [7].

Participating in the World Trade Organization (WTO) constitutes a competitive advantage that enables countries to access high-value markets. By promoting free trade by reducing tariffs in global markets, member countries have the ability to compete with other exporters and domestic products. This is an important factor that influences price and enhances the national shrimp industry.

Shrimp disease is a challenge for producer countries because it threatens the volume and quality of the exported product, and is therefore a competitive disadvantage. Previous studies have evaluated losses caused by early mortality syndrome on commercial shrimp aquaculture [8], economic losses [9], and the global perspective of shrimp disease [10]. The losses due to EMS in Vietnam have been reported [11–13].

The application of computing techniques for farming enhancement, disease prediction, and market trend analysis is widely used. For example, an expert system [14] and tools for processing digital images [15] were used to diagnose shrimp disease. Machine learning has been used for predicting disease occurrence in cultured shrimp. Previous studies used machine learning to predict the occurrence of shrimp disease [16–18] and create applications for aquaculture [19]. Machine learning has also been used in sales forecasting [20–22]. Researchers [23] applied a regression model to predict the stock price, while other research [24] used a random forest algorithm to forecast supply chain demand.

Accurate price prediction is very important for fishery exports because it helps to determine global market trends, enhancing the quality of seafood products. There are millions of tons of shrimp exported to the international market from producer countries and, therefore, estimates of price need to be as precise as possible. This information can be used by exporters to determine strategies to increase exports, leading to more financial benefits for national economies and providing motivation to shrimp farmers.

Although machine learning algorithms are useful for making predictions, they still depend on accurate datasets and each algorithm has its own strength of prediction. For example, the

random forest algorithm was outperformed in terms of prediction by a dataset of power generation and power system security [25], and a logistic regression performed better than a neural network in the prediction of occurrence of early mortality syndrome [17]. A study preferred a probabilistic neural network to a logistic regression model when analyzing a dataset of general shrimp diseases [16]. To increase accuracy and overcome the dependence of the algorithm on the dataset, the combination of many machine learning algorithms was proposed [26]. This enabled the generation of a more powerful predictive model, called the super learner. Researchers [27] used the super learner to predict the phenotypic antiretroviral susceptibility of HIV in humans and found that it was as good as or better than any single algorithm. A combination of ten single machine learning algorithms (a neural network, linear regression, randomized trees, XGBoost, loess, random forest, polyMARS, MARS, lasso, and support vector regression) [28], was used to optimize the accuracy of daily stream flow forecasts. One study [29] used the super learner to improve the accuracy of prediction of mortality risk in an older population.

In addition to improving accuracy, it is also important to determine the importance of each variable used in predictions. For exports, this will help identify the factors that have advantageous and disadvantageous influences on the price of products. The producer will then develop strategies to boost exports by increasing the beneficial conditions and minimizing the negative effects. However, there are difficulties with the machine learning approach because it creates a black-box model that is difficult to interpret, making it difficult for humans to understand and trust.

Due to the effort required to evaluate the contribution of each variable in the output of a machine learning analysis, the SHapley Additive exPlanations (SHAP) method was introduced to interpret predictive models. In 2017, a unified approach used SHAP as a way to explain the importance of each variable in the output of a machine learning analysis [30]. An interpretation by SHAP was also conducted in a previous study [31]. SHAP not only successfully explained the internal logic of prediction but also verified the credibility of a predictive model [32].

In this study, we used a super learner, which had the potential to give accurate and stable price predictions for Vietnamese shrimp products exported to the US market. It combined 10 candidate algorithms to predict the exported price of Vietnamese shrimp based on information from competitive exporters (China, Thailand, Indonesia, India, Ecuador, and Chile). To interpret the prediction, SHAP was used to determine how each predictor influenced the export price and then suggested solutions for the development of the Vietnamese shrimp industry. This study provided a new approach to make accurate Vietnamese price predictions based on information from competitors rather than the use of information solely from Vietnam, as in previous research [33]. It also provided a method to interpret the predicted result.

## Materials and methods

### Preliminaries

Monthly data was collected from the US Department of Agriculture, WTO, and FAO for the period from May-1995 to May-2019. The seven leading exporters of frozen shrimp products to the US market (China, Thailand, India, Indonesia, Ecuador, Chile, and Vietnam) were included in the dataset. The exporting countries were direct competitors in the US market. Therefore, an increase in the price of exports from any of the listed countries would shift the demand curve of shrimp products imported to the US market [34]. This would lead to an increase in demand for shrimp products imported from other countries among US consumers. The prices of shrimp products imported from other countries would then increase.

Additionally, the variables influencing the Vietnamese export price, and the competitive advantages and drawbacks (i.e., shrimp disease) of these countries were assessed. The difference in price between Vietnam and each competitor country was considered. The correlations between the variables and price in Vietnam and other countries were evaluated and it was determined whether they were negative or positive. A negative value meant that the Vietnamese price was lower than that of the other countries and vice versa. The export price was presented in US dollars; thus, the correlated price was also reported in US dollars. In Fig 1, the correlations between Vietnamese export prices and those of other countries are shown. The prices of Vietnamese exported shrimp were mainly found to be between 10–15 USD per kg, while prices >15 and <10 USD were no common. Export prices of other countries had a wider but smaller distribution range.

The US is one of the largest markets for seafood products in the world. Product quality, food safety, and traceability issues are extremely important in the US market. Shrimp producers that export to the US need to adhere strictly to product requirements. We selected some of the mandatory US requirements that applied to shrimp products from all producer countries, such as global GAP, Hazard Analysis Critical Control Point, Safe Quality Food, and Aquaculture Steward Council. Global GAP and Safe Quality Food aim to provide a food safety assurance that protects consumer heath, while Hazard Analysis Critical Control Point guarantees the product origin. The Aquaculture Steward Council practices are used to enhance the responsibility of producers in terms of minimizing their impact on the environment. This certificate requires a limitation on the use of wild fish as an ingredient in shrimp feed, as well as the regular assessment of water quality to avoid pollution and disease outbreaks. These farming certificates are not only needed in response to the requirements of the US market but will also beneficially increase the price of products and the competitive advantages among exporters in the long term due to improvements in product quality [35]. The implementation date for the certificates differs among the producer countries, i.e., Vietnam has applied global GAP since Sep-2007, while Indonesia has only applied this certification to shrimp since Oct-2011.

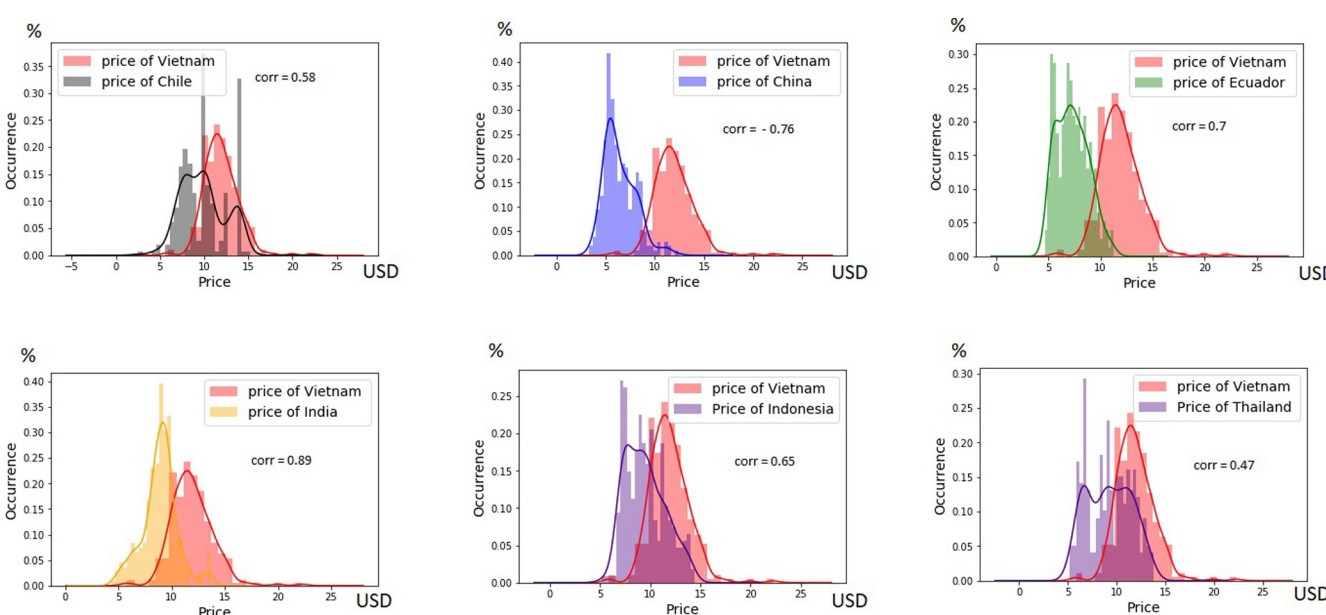

**Fig 1. The correlations price between Vietnamese export shrimp and that of other countries.**

This was considered likely to affect the export price as both countries export to the same destination, i.e., the US market.

To protect the domestic shrimp industry, anti-dumping tariffs were set by the US government in 2003. These apply to China, India, Thailand, Vietnam, and Ecuador, while Indonesia and Chile are not affected by this requirement [7]. It was considered that the Vietnamese price was impacted by this practice, leading to an inability to compete with the non-impacted countries. To participate in global trade, many countries have attempted to become members of the WTO. This would provide an opportunity to access stringent markets. The different times at which producer countries have joined the WTO are likely to have affected the export price. The number of shrimp exporting countries participating in the WTO will affect the price of Vietnam's shrimp export. Membership of the WTO was therefore selected as one of variables in the super learner process to evaluate the export price.

Disease is a serious concern in export shrimp production. It can cause problems for producer countries due to US market concerns regarding disease transmission and residual chemicals in the final product due to the materials applied for treatment. This will reduce the competitiveness of exporters and export volumes. Before imports can be accepted, shrimp producers in countries where disease is confirmed could be required to show evidence of safe products. Therefore, early mortality syndrome was selected as a variable that could influence the Vietnamese export price.

Other variables in the dataset, such as enrofloxacin antibiotic residues and Country of Origin Labeling were not used in predictions. Because they applied to all export countries at the same time, they were not meaningful for predicting price variations. We only focused on competitor information and, therefore, the exchange rate and US income per capita were also omitted. To validate the association between independent variables that were used to hypothesize the target value–Vietnamese export price, the Pearson correlation [36] method was applied. This method measures the strength of the relationship between two variables [37], based on their coefficient. The Pearson correlation coefficient between variable X and Y is defined as:

$$R = \frac{cov(X, Y)}{\sqrt{var(X)var(Y)}} \tag{1}$$

where, *cov* is the covariance and *var* is the variance. Variables that had a high correlation with the Vietnamese export price were selected for prediction. We obtained 13 independent variables for use in the super learner process to predict the Vietnamese export price (see Table 1).

**Linear regression.** Linear regression builds models that assume a linear relationship between input variables (x) and the single output variable (y) via scale factors to each input, called coefficients (β). The formula of this algorithm is as follows:

$$y_i = \beta_0 + \beta_1 x_{i1} + \ldots + \beta_p x_{ip} + \varepsilon \tag{2}$$

where $\beta_0$ is the intercept (when x is 0), i indicates the $i^{th}$ sample in the dataset, $\varepsilon$ is a random variable, x corresponds to variables in the dataset, p is the $p^{th}$ independent variable, and y is the Vietnamese export price.

Parameters were set to increase the reliability of the predictions for this algorithm. In the scikit-learn Python package, parameter *fit_intercept*, which is $\beta_0$ in (2), was set to "True" to calculate the intercept for this model. *Normalize* and *copy_X*, which are used to normalize by L2-norm and copy all variables in the dataset, were set to "True." Other parameters such as *n-jobs* and *positive* were set to default.

**Table 1. List of variables.**

| Variables | Description |
|---|---|
| DifferencePriceVN_China | Price gap between Vietnam and China |
| DifferencePriceVN_Indonesia | Price gap between Vietnam and Indonesia |
| DifferencePriceVN_India | Price gap between Vietnam and India |
| DifferencePriceVN_Thailand | Price gap between Vietnam and Thailand |
| DifferencePriceVN_Chile | Price gap between Vietnam and Chile |
| DifferencePriceVN_Ecuador | Price gap between Vietnam and Ecuador |
| Certificated_SQF | Number of competitive countries with Safe Quality Food certificates |
| Certificated_HACCP | Number of competitive countries with Hazard Analysis and Critical Control Point certifications |
| Certificated_ASC | Number of competitive countries with Aquaculture Stewardship Council certificates |
| Infected_EMS | Number of competitive countries confirmed to be infected by Early Mortality Symptom in cultured shrimp |
| Member_WTO | Number of competitive countries that are members of the WTO |
| Applied_GAP | Number of competitive countries that apply global Good Agricultural Practice |
| Imposed_ANTI | Number of competitive countries subject to anti-dumping laws by the USA. |

**Lasso regression.** The lasso regression, which is an abbreviation of "least absolute shrinkage and selection operator," is used for variable selection and estimation in linear regression models [38]. A constraint is imposed on model parameters to make regression coefficients for variables shrink to zero. Some variables have a zero-coefficient and are eliminated, while non-zero coefficient variables are used to evaluate the model. The coefficient of linear regression ($\beta_0$, $\beta_1$,...,$\beta_P$) in lasso is as follows:

$$\widehat{\beta}_\lambda = argmin \, \|y - x\beta\|_2^2 + \lambda_L \|\beta\|_1 \tag{3}$$

where, x and y are input and output variables, respectively, and $\lambda$ is a non-negative tuning parameter that is used to control the shrinkage. The higher the value of $\lambda$, the larger the shrinkage of the model.

Similar to linear regression, the parameters in this algorithm including *fit_intercept*, *normalize*, and *copy_X* were set to "True." The parameter *alpha* was used to control regularization strength with values at 0.8 being the best. *Max-inter*, the maximum number of iterations, was set to 500. Other parameters such as *tol* (used for stopping criterion), *precompute*, *random_state*, and others were given default values.

**Ridge regression.** This is a bias estimation method that is used for estimating the coefficients of a regression model where the independent variables are strongly correlated [39]. The term bias in machine learning can be understood as the extent to which the model fails to produce a plot that is in line with the samples. The ridge regression imposes a penalty term to coefficient $\beta$ to control bias, thus improving the accuracy of prediction. The cost function for a ridge regression performs an L2 regulation as follows:

$$f(\beta) = min(\|y - x(\beta)\|_2^2 + \lambda \|\beta\|_2^2) \tag{4}$$

where, $\lambda$ is a penalty term, and x and y are the input and output variables, respectively.

To fairly evaluate ridge, linear, and lasso regression, the parameters of ridge including *fit_intercept*, *normalize*, and *copy_X* were also set to "True." The *max_iter* was set to 500 and alpha was set to 0.8. The *tol*, *positive*, and *random_state* were set to defaults. The parameter

*solver* was set to "svd" which is singular value decomposition of independent data (13 independent variables in the dataset) to compute the Ridge coefficients.

**Elastic net.** Elastic net regression uses penalties from both the lasso and ridge methods to regularize a regression model. It improves the regularization of the predicted model by learning the shortcomings of the ridge and lasso models. The limitation of the lasso regression is that it used only a few samples to produce high dimensional data (too many variables), while the ridge method can keep many highly correlated variables in the dataset. To overcome these issues, the elastic net performs a variable selection and regularization simultaneously. Accordingly, there are two stages involving the lasso and ridge methods: first, it finds the ridge regression coefficient and, second, it uses a lasso-type shrinkage of the coefficient.

To implement this algorithm, we set parameter values the same as the ridge algorithm: *alpha* equal to 0.8; *max_inter* at 500; and *fit_intercept*, *normalize*, and *copy_X* set to "True." Here, the parameter *l1_ration* was set to 0.5 and *selection*, which was used to update the coefficient in every iteration was set to the "random" option.

**K-nearest neighbor.** This method is based on the similarity concept that assumes similar things exist in close proximity [40]. Hence, k-nearest neighbor finds the similarity among data points by calculating the distance among them. The Euclidean distance is often used to measure how close two data points are. Here, K represents the specified number of samples that need to be grouped in terms of similarity. All the samples in one group have the same label (or value). Therefore, once a new sample is placed into a specific group, it is assigned the label of that group. Depending on the data, an appropriate k-value must be established. In this study, we took $K = 5$ to make predictions. The parameter *weights* was set to the value "distance" to indicate that closer neighbors of a query point will have a greater influence than neighbors that are farther away. *Leaf_size* was set to 20, the distance *metric* was set to "minkowski," and *algorithm* was set to "kd_tree." Other parameters were set to default values.

**Support vector regression.** A support vector machine attempts to determine a line (called the hyperplane in multidimensional space) that will separate two or many classes of data. Support vector regression was built on the principle of a support vector machine, but is used for regression problems [41]. A support vector regression performs the mapping between inputs and outputs by developing a hyperplane. The data points on either side of the hyperplane that are closest to the hyperplane are called support vectors and are used to determine the boundary line. To increase the prediction accuracy, a support vector regression attempts to find the best hyperplane within a threshold value (distance between the hyperplane and boundary line), instead of minimizing the error between the real and predicted values.

To implement this algorithm, the parameter *kernel* was set to "sigmoid"; *degree* was set to 3; *gamma* was set to "auto"; *cache_size*, used to specify the size of the kernel cache, was set to 100 megabytes; *max_iter* was set to 500. For the regularization parameter *C*, which is known to be a penalty parameter of the error term, was set to 2. Other parameters such as *gamma*, *tol*, *shrinking*, and *epsilon* were set to default values.

**Decision tree.** The decision tree algorithm is based on the structure of a tree to predict the outcome from independent variables in both classification and regression. There are many nodes located from the root to the branches of the tree. In this architecture, the root node and internal nodes are labeled with input values, while the leaf is the output value. Depending on the type of output value, a decision tree is used for classification or regression. A decision tree where the target variable can take continuous values is called a regression tree. Here, we used a regression tree to predict the Vietnamese export price.

In this algorithm, the parameter *criterion* used to measure the quality of a split brand of tree, was set to "gini." *Splitter* strategy was set to "random." *Max_depth* was set to 10.

*Min_samples_slit* was set to 5. *Min_samples_leaf* was set to 5. Other parameters such as *min_-weight_fraction_leaf*, *max_features*, and *random_state* were set to default values.

**Random forest.** The random forest is based on the decision tree concept [42]. As its name suggests, multiple trees are built to make a prediction. The random forest algorithm randomly selects samples and uses the best split of a subset of variables to build, simultaneously, multiple sub-decision trees. Majority voting is used to obtain the final result, which is then applied to the sub-decision trees. The random forest is therefore more flexible than a decision tree.

In the implementation of random forest, the *n_estimator*, used to indicate the number of trees, was set to 1,000. The parameter *criterion* function, used to measure the quality of a split in the forest, was set to "entropy." The parameter *min_samples_split*, which indicates the minimum number of samples required to split an internal node, was set to 5. The *max_depth* was set to 5. Other parameters, such as *min_samples_leaf*, *min_weight_fraction_leaf*, *max_features*, *max_features*, and *min_impurity_split* were set to the default values.

**Gradient boosting.** Gradient boosting is also based on the decision tree concept. Unlike the decision tree, gradient boosting sequentially builds sub-decision trees. The next sub-tree is built with the purpose of improving the error of the previous tree to enhance the ensemble performance [43]. The process continues to build sub-trees until the specified number of iterations is reached. The prediction of the final model is the sum of the predictions of previous tree models.

Similar to random forest, *n_estimators* was set to 1,000 in the implementation of gradient boosting algorithm. The parameter *max_depth* was set to 5. The *loss* function used was least squares regression. The parameter *min_samples_split* was also set to 5. The parameter *subsample* used to control variance and bias was set equal to 1. Other parameters, such as *alpha*, *max_-features*, and *min_impurity_split*, were set to their default values.

**Neural network.** This algorithm is inspired by the human brain [44]. There are many connected nodes located inside multiple layers in the structure of this algorithm. The computation is performed inside each node and provides an output by mathematically processing inputs with connected weights. The previous nodes will output a value, which is then used as the input for the next node in the network. Generally, the complex structure of a neural network consists of multiple hidden layers between the input and output layers. The complexity of the algorithm is generated through a number of hidden layers that are used for mapping the input and output. In this study, we applied a neural network with five layers: one input layer to receive values from the independent variable, three hidden layers for the mapping process, and one output layer for the target variable.

For the neural network, the multi-layer perceptron regressor of scikit-learn was used to obtain the prediction. The parameter *hidden_layer_size* was set to 3 layers, each with 30 neurons. The *activation* fuction was set to "tanh." The *solver* was set to "lbfgs." The *batch_size*, which identifies the size of minibatches for stochastic optimizers, was set to 10. The *learning_rate*, used to schedule for weight updates, was set to "constant." Other parameters were set to default values.

**Extra tree regression.** Extra tree regression is an ensemble technique that works by creating a large number of unpruned sub-decision trees from the training dataset. The extra tree prediction is made by averaging the predictions of the sub-decision trees. This technique is similar to that of the random forest, but it randomly chooses a subset of features to build sub-trees, whereas the random forest makes the optimal choice. The difference makes the extra tree run faster because it does not need to calculate the optimal pathway. In this study, the extra tree algorithm is used to combine the predictions of the 10 candidate algorithms and make the final prediction in the super learner process.

Extra tree was implemented to obtain the final prediction for the super learner with specific parameters. The *n_estimators* was set to 1,000. The parameter *max_depth* was set to 5. The *criterion* was set to "absolute error." The parameter *min_samples_split* also was set to 5. Other parameters, such as *min_weight_fraction_leaf*, *bootstrap*, and *min_impurity_decrease*, were set to their default values.

In this study, all of the algorithms used were obtained from the scikit-learn package [45], which is supported by Python scripts.

## Methodology

The super learner is a prediction method that allows researchers to combine the results of a set of single machine learning algorithms into one to improve the predictive performance [46]. The advantage of this method is that the prediction accuracy of its model is as good as or better than any model from a single algorithm. This method is based on the theory of cross-validation and can generate the optimal weighted combination among base algorithms, which is both adaptive and robust for use with a small number of samples [47]. The cross-validation means that all candidate models used the same k-fold splits in the dataset. Due to the super learner being developed using a stacked generalization technique, it uses a new model to combine the predictions from multiple candidate models that are already trained. To predict the accuracy of Vietnamese export prices, random forest and gradient boosting were selected as the best single machine learning approaches [33]. Therefore, these two algorithms were first chosen for use in the super learner. Forward selection was applied to iteratively add new potential candidate algorithms to the super learner. To be selected into the model, the potential algorithm had to contribute to the super learner and reduce the error of model. The more algorithms added, the more accurate the ensemble model. However, too many candidate algorithms will increase the time and computer cost for implementation. To balance the accuracy and computer cost, we set the number of algorithms in the super learner to 10. The forward selection settings for predicting the export price is described in Table 2.

After evaluating the suitability of the combination of algorithms, we used the above 10 candidate algorithms (linear regression, elastic net, k-nearest neighbor, support vector regression, decision tree, random forest, gradient boosting, neural network, lasso, and ridge) to make the base prediction. Then the extra trees algorithm was used to combine all base predictions and obtain the final result. The concept of the super learner is shown in Fig 2. Here, we divided the dataset into two subsets: 75% and 25% for the training and testing sets, respectively. According to a time series analysis, the training set used the data for the period from May-1995 to Apr-2013, while the testing set consisted of the following time period from May-2013 to May-2019.

**Table 2. Forward selection of candidate algorithms.**

| Step | Candidate algorithm | MAPE |
|:---:|---|:---:|
| 0 | Random forest, Gradient boosting | 5.16% |
| 1 | Elastic net | 3.21% |
| 2 | Lasso | 2.84% |
| 3 | Decision tree | 2.27% |
| 4 | K-nearest neighbor | 1.95% |
| 5 | Linear regression | 1.16% |
| 6 | SVR | 1.01% |
| 7 | Bridge | 0.95% |
| 8 | Neural network | 0.80% |

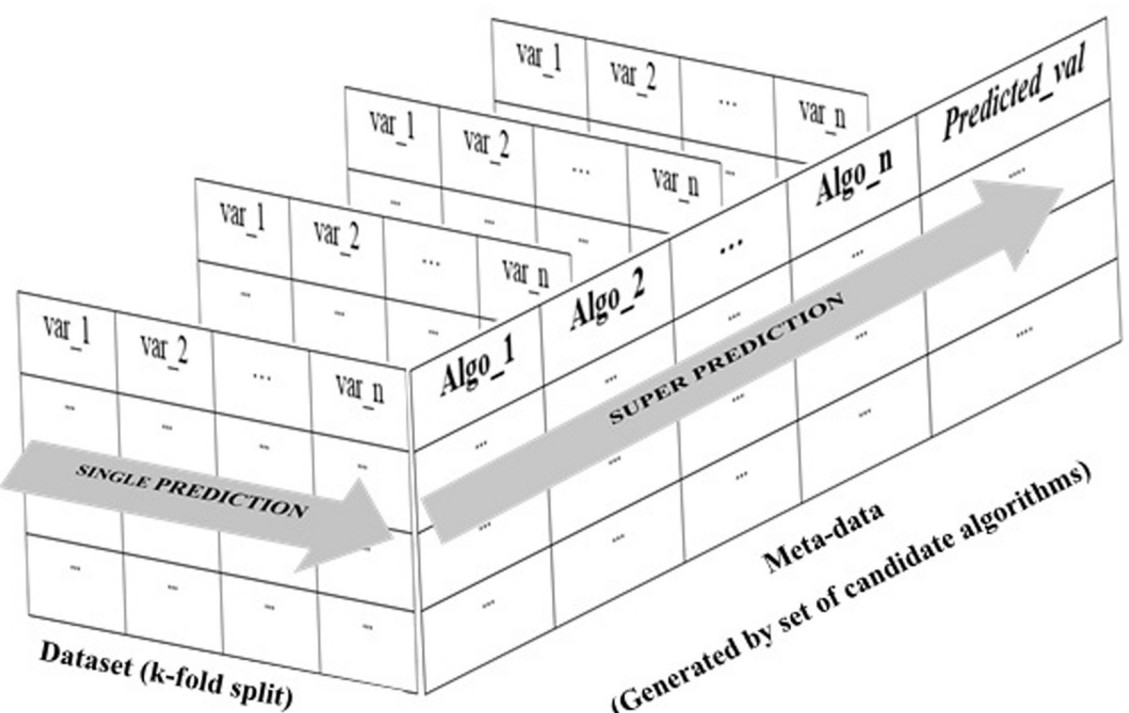

**Fig 2. Concept of the super learner.** Candidate algorithm (Algo_1 to Algo_n) and independent variable in dataset (var_1 to var_n).

To determine how each variable influenced the price, SHAP is used. Originally, SHAP was developed by Shapley in 1953 [48] to estimate the importance of an individual player in a collaborative team game. It evaluates the contribution among players to the final result of the game. It is based on an optimal Shapley value. The Shapley value is calculated by averaging the marginal contribution of the variable's values across all possible sets (coalitions). Thus, the Shapley value indicates how to distribute the predictions fairly among independent variables. This concept was later developed to interpret the contribution of each predictor in a machine learning analysis [49]. A prediction can therefore be made by assuming that an independent variable of all points in a dataset acts as a "player" in game and the predicted result is the payout [48].

Accordingly, SHAP is used to evaluate the importance of variables for an outcome when making predictions [50]. The contribution of each variable $i$ is indicated by the SHAP value as follows:

$$\Phi_i = \frac{1}{|N|!}\sum_{S \subseteq N|\{i\}} |S|!(|N| - |S| - 1)! \left[f(S \cup \{i\}) - f(S)\right] \tag{5}$$

where $f(S)$ is the outcome of the sub-set variable $S$ used in a machine learning model, while N is the complete set of all variables. The contribution of variable $i$ (called $\Phi_i$) could have negative and positive signs. A positive value contributes to the prediction of activity, while a negative value contributes to the prediction of inactivity. In this method, the difference between the average and component prediction (for each subset $S$) is fairly distributed among the variables of interest, which guarantees that a full explanation of the predicted result will be delivered. Therefore, SHAP is an ideal solution in the interpretation of prediction problems. However, the computing power required to estimate the Shapley value is extremely large because it needs to handle $2^k$ possible sets of variable values. In the exponential number of subsets, the $k$

number is found by sampling sets and the number of iterations (called *M*). There is no rule for setting the optimal *M* due to a decrease in *M* reducing the computing time, leading to an increase in the variance of the Shapley value and vice versa. Hence, the value of *M* needs to be large enough to obtain a good Shapley value and small enough to complete the computation in reasonable time [49]. In this study, we used the SHAP package, supported by python, to estimate the Shapley value for each variable in the dataset.

To visualize the contribution of each variable, a summary plot was constructed. Each point was a Shapley value for a variable and a data point. The y-axis is a set of variables, the x-axis is a Shapley value, and the color indicates the value of the variable from low to high. In the summary plot, the importance of variables is ranked in descending order, enabling us to determine which variable is the most important and which is the worst. We evaluated the contribution of all of the independent variables related to competitive factors for the prediction of export price.

## Results

### Variable selection

The Pearson correlation method was applied to find the variable with the strongest association with the Vietnamese export price. The correlation assigns a value between −1 and 1, where 0 is no correlation, 1 is the total positive correlation, and −1 is the total negative correlation. For continuous variables including differencePriceVN_China, differencePriceVN_Chile, differencePriceVN_Thailand, differencePriceVN_Indonesia, differencePriceVN_India and differencePrice_VN_Ecuador, the correlation values were -0.76, 0.58, 0.47, 0.65, 0.89 and 0.72, respectively. For binary variables such as certificated_SQF, Certificated_HACCP, Certificated_ASC, Infected_EMS, Member_WTO, Applied_GAP, Imposed_ANTI, we found the high correlation values with Vietnamese price which ranges from 0.52 to 0.96 of correlation. These high correlation variables were used to hypothesize the Vietnamese export price for 3-, 6-, 9-, and 12-month base predictions.

### Prediction accuracy

We predicted the export price based on historical data for the previous 3, 6, 9, and 12 months. We used the mean absolute error (MAE), and mean squared error (MSE) to measure the accuracy of prediction in the testing subset (period from May-2013 to May-2019) as follows:

$$MAE = \frac{1}{m} \sum_{i=1}^{m} |y_i - \widehat{y}_i| \tag{6}$$

$$MSE = \frac{1}{m} \sum_{i=1}^{m} (y_i - \widehat{y}_i)^2 \tag{7}$$

where *m* is the number of test samples, $y_i$ is the actual value, and $\widehat{y}_i$ is the predicted value.

Then, the percentage error was calculated (for MAE, mean absolute percentage error MAPE = (MAE × 100)/average price, and for MSE, mean square percentage error MSPE = (MSE × 100)/average price $^2$). The average price in the dataset was 11.9 USD. The results obtained using the candidate algorithms and super learner are given in Table 3.

In the 3 month period, the super learner produced a prediction result with a MAPE of 0.8% and MSPE of 0.01%. This was an improvement on the accuracy of the single algorithms. The average MAPE of the candidate algorithms was 6.81%, while the MSPE was 0.83%. Accordingly, the super learner method improved the MAPE by about 6 percentage points and the MSPE by about 0.80 percentage points compared to the single algorithm approach. Among the candidate algorithms, the best performance was achieved by lasso (MAPE of 5.46%, MSPE

**Table 3. Prediction for 3, 6, 9, and 12 months base.**

| Period base | | Candidate algorithm | | | | | | | | | | Supper learning |
|---|---|---|---|---|---|---|---|---|---|---|---|---|
| | | Linear Reg. | Elastic Net | SVR | Decision Tree | K-NN | Random Forest | Gradient Boosting | Neural network | Ridge | Lasso | |
| 3 months | MAE | 0.822 | 0.755 | 0.696 | 1.036 | 0.794 | 0.788 | 0.708 | 1.195 | 0.663 | 0.650 | 0.095 |
| | MAPE | 6.91% | 6.34% | 5.85% | 8.71% | 6.67% | 6.62% | 5.95% | 10.04% | 5.57% | 5.46% | 0.80% |
| | MSE | 1.063 | 0.867 | 0.677 | 1.783 | 0.973 | 0.978 | 0.782 | 3.386 | 0.637 | 0.651 | 0.021 |
| | MSPE | 0.75% | 0.61% | 0.48% | 1.26% | 0.69% | 0.69% | 0.55% | 2.39% | 0.45% | 0.46% | 0.01% |
| 6 months | MAE | 1.156 | 0.761 | 0.717 | 0.893 | 0.781 | 0.719 | 0.734 | 1.644 | 1.037 | 0.705 | 0.142 |
| | MAPE | 9.71% | 6.39% | 6.03% | 7.50% | 6.56% | 6.04% | 6.17% | 13.82% | 8.71% | 5.92% | 1.19% |
| | MSE | 2.236 | 0.872 | 0.714 | 1.313 | 1.016 | 0.761 | 0.761 | 5.655 | 1.777 | 0.701 | 0.063 |
| | MSPE | 1.58% | 0.62% | 0.50% | 0.93% | 0.72% | 0.54% | 0.54% | 3.99% | 1.25% | 0.50% | 0.04% |
| 9 months | MAE | 2.047 | 0.748 | 0.720 | 0.742 | 0.835 | 0.707 | 0.696 | 1.509 | 1.510 | 0.668 | 0.126 |
| | MAPE | 17.20% | 6.29% | 6.05% | 6.24% | 7.02% | 5.94% | 5.85% | 12.68% | 12.69% | 5.61% | 1.06% |
| | MSE | 6.358 | 0.875 | 0.747 | 0.956 | 1.199 | 0.770 | 0.729 | 4.103 | 3.549 | 0.705 | 0.044 |
| | MSPE | 4.49% | 0.62% | 0.53% | 0.68% | 0.85% | 0.54% | 0.51% | 2.90% | 2.51% | 0.50% | 0.03% |
| 12 months | MAE | 3.556 | 0.750 | 0.783 | 1.033 | 0.955 | 0.867 | 0.755 | 2.714 | 1.992 | 0.704 | 0.133 |
| | MAPE | 29.88% | 6.30% | 6.58% | 8.68% | 8.03% | 7.29% | 6.34% | 22.81% | 16.74% | 5.92% | 1.12% |
| | MSE | 19.157 | 0.889 | 0.878 | 1.714 | 1.432 | 1.222 | 0.846 | 9.880 | 6.254 | 0.772 | 0.032 |
| | MSPE | 13.53% | 0.63% | 0.62% | 1.21% | 1.01% | 0.86% | 0.60% | 6.98% | 4.42% | 0.55% | 0.02% |

of 0.46%) and the ridge method (MAPE of 5.57% and MSPE of 0.45%), while the worst accuracy was achieved by the neural network (MAPE of 10.04% and MSPE of 2.39%). Fig 3A shows how close the predicted values from the super learner were to actual values.

In the 6-month period, the super learner produced a prediction result with a MAPE of 1.19% and MSPE of 0.04%. The combined approach substantially reduced the error compared to the average prediction from the candidate algorithms (MAPE of 6.50% and MSPE of 1.08%). Among the candidate algorithms, lasso achieved the highest accuracy (MAPE of 5.92% and MSPE of 0.50%), while the neural network (MAPE of 13.82% and MSPE of 3.99%) had the worst performance. Fig 3B shows the prediction for this 6-month period.

In the 9-month period, the super learner produced a prediction result with a MAPE of 1.06% and MSPE of 0.03%. Among the candidate algorithms, lasso achieved the highest accuracy (MAPE of 5.61% and MSPE of 0.50%), while linear regression produced the worst prediction with the highest error (MAPE of 17.02% and MSPE of 4.49%). Compared to the best candidate algorithm (lasso), the super method improved the error by at least 4 percentage points (for MAPE) and 0.47 percentage points (for MSPE). The accuracy of the predictions is presented in Fig 3C.

In the 12-month period, the super learner produced a prediction result with a MAPE of 1.12% and MSPE of 0.02%. In the stand-alone approach, the accuracy of some candidate algorithms was substantially different. The lowest MAPE and MSPE were obtained using lasso (5.92% and 0.55%, respectively). The MAPE values obtained with the ridge, neural network, and linear regression methods were large (16.74%, 22.81%, and 19.88%, respectively), while the other algorithms produced predictions with MAPE values in the range of 6.03–8.68%. Fig 3D shows a comparison of the actual and predicted values of the super learner.

## The SHAP evaluation

To indicate the importance of each variable used in the dataset, a summary plot of SHAP values is presented in Fig 4. These contributions were ranked in descending order, with the

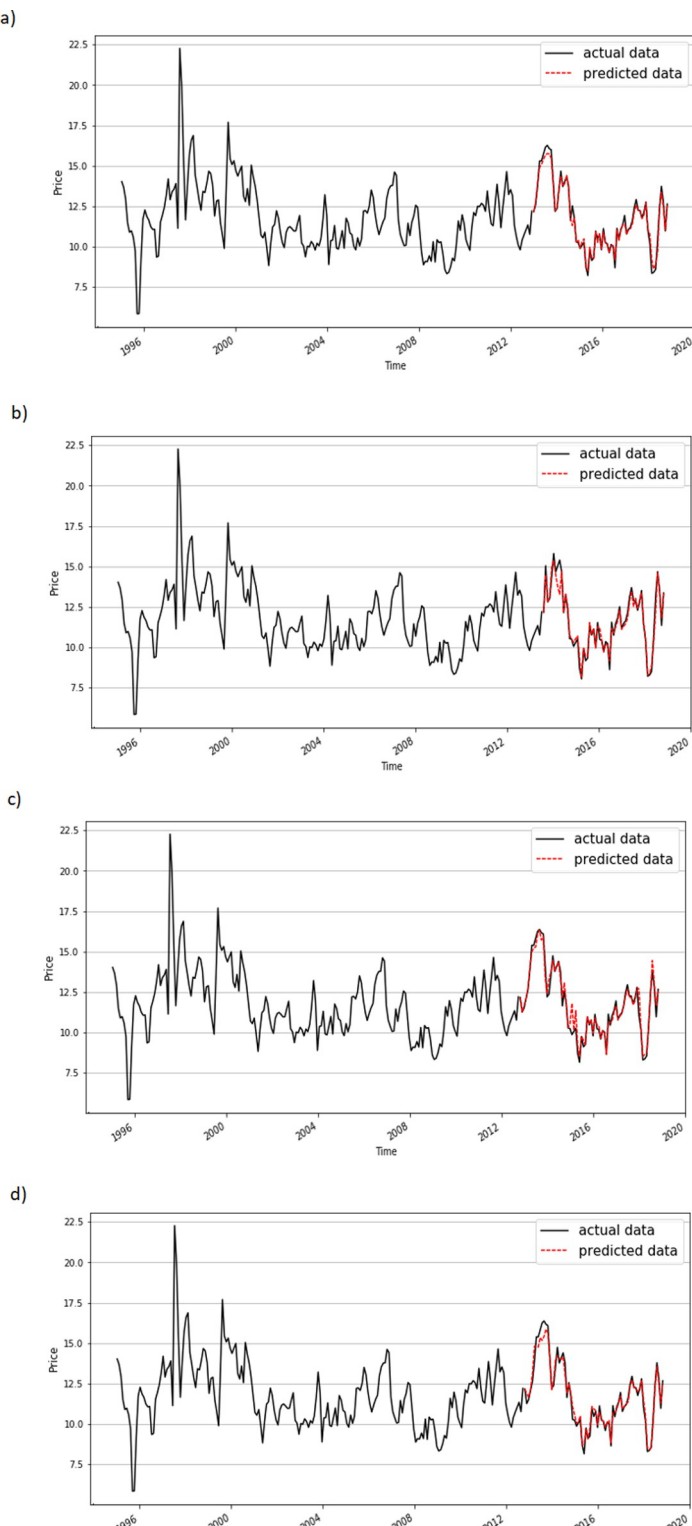

**Fig 3.**  a. Prediction for a 3-month base by the super learner. b. Prediction for a 6-month base by the super learner. c. Prediction for a 9-month base by the super learner. d. Prediction for 12-month base by the super learner.

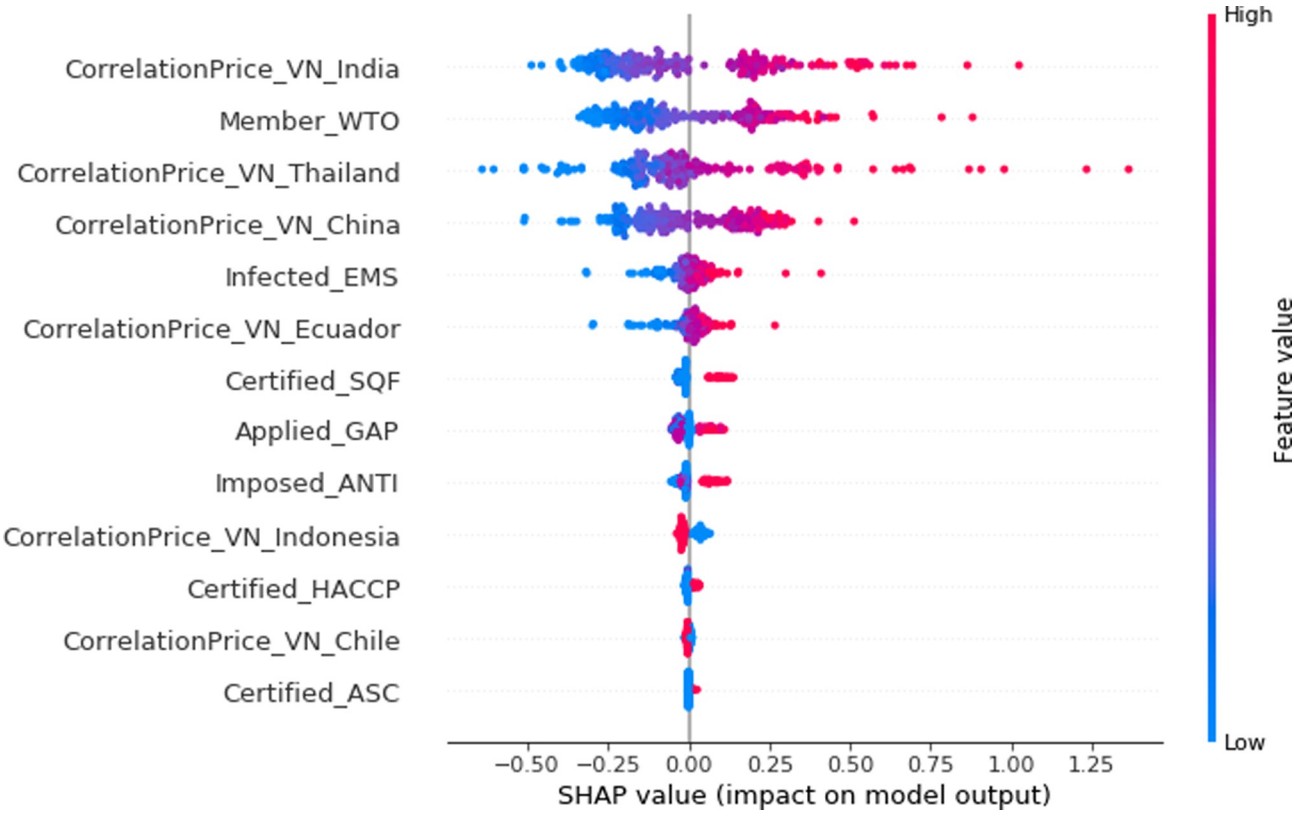

**Fig 4. SHAP interpretation.**

highest importance at the top and the lowest at the bottom. The horizontal location in each feature indicates its impact on the prediction, i.e., red had a strong influence on prediction, while blue had a weak influence. The results indicated that differencePriceVN_India, member_WTO, differencePriceVN_Thailand, differencePriceVN_China, infected_EMS and differencePriceVN_Ecuador had large and positive impacts on the prediction of the target variable (Vietnamese export price). The strong influence was indicated by the red color and there was a positive impact indicated on the right axis of the SHAP figure. Generally, the difference in export price between Vietnam and other competitive exporters had a strong influence on the predictive model, while farming certificates had little impact on the accuracy. Among the competitors, India had the most influence on the Vietnamese export price. Participating as a member of WTO was the second most important factor, followed by Thailand, China, outbreak of disease (early mortality syndrome), and Ecuador, while Indonesia, Chile, and farming certificates (Safe Quality Food, Hazard Analysis Critical Control Point, and Aquaculture Steward Council) were less significant.

## Discussion

The super learner produced predictions that not only obtained a high accuracy but were also stable for different periods of historical data. The MAPE in all predictions was very low at 0.8%, 1.19%, 1.06%, and 1.12% for the 3-, 6-, 9-, and 12-month periods, respectively. Similarly to MAPE, the optimal MSPE was obtained with the super learner, with values of 0.01% (3 months), 0.04% (6 months), 0.03% (9 months), and 0.02% (12 months). Compared to the performance of each candidate in the base algorithms, this method completely improved the error

of prediction. The MAPE was improved by more than 4 percentage points using the super learner compared to the best candidate algorithm. It significantly reduced the error from the best single candidate (lasso), with reductions from 5.46% to 0.8%, 5.92% to 1.19%, 5.61% to 1.06%, and 5.92% to 1.12% for the 3-, 6-, 9-, and 12-month base predictions, respectively. There was an improvement of at least 0.4 percentage points in the MSPE for the super learner compared to the best single approach. Additionally, the combined method resulted in a stable prediction, meaning that the accuracy was less dependent on historical data. The MAPE values for the 6 and 12 month periods were 1.12% and 1.19%, respectively. Fig 3 shows the stable accuracy of the super learner, which minimized errors in all predictions. These data prove that the super learner is a suitable approach for predicting the Vietnamese price of export shrimp. The combination of candidate algorithms makes the ensemble model powerful, overcoming the dependence of each algorithm on the dataset.

This work enhanced the previous predictions presented by [33]. A high accuracy and stable prediction were obtained using the super learner, which outperformed the single approaches of the random forest and gradient boosting. To gain an advantage in a global market, a producer not only has to satisfy the mandatory product requirements but also needs to reach an agreement on an appropriate price. Therefore, accurately predicting the price of an exported shrimp product is essential for enabling a producer to compete in the market.

The factors influencing the prediction were evaluated. In the SHAP (shown in Fig 4), the variable with the largest impact was the correlation in price between Vietnam and India. This was because India is the country exporting the most frozen shrimp in the global market (3.89 billion USD in 2019), followed by Ecuador, Vietnam, and Indonesia at 3.6 billion USD, 1.9 billion USD, and 1.4 billion USD, respectively. About 32% of all shrimp imported to the US originates from India [51], and it therefore has a very strong competitive position, significantly affecting other exporters, including Vietnam.

In our analysis, the correlation between Indian and Vietnamese prices (in Fig 1) had an associate value of 0.89, the highest in our study. The distribution of Indian prices was similar to that of Vietnamese prices but lower. Indian prices ranged mainly between 8 and 11 USD, while Vietnam prices were 10–15 USD. This means that India has lower export prices, leading to a competitive advantage with Vietnam estimated in the SHAP evaluation (Fig 4).

India has promoted the development of shrimp production, which has become a key sector of the Indian economy, contributing 70% of the value of India's seafood exports [52]. From 2011 to 2018, farmed shrimp production in India increased by 23%, far surpassing the average global growth rate of 5.6% [53].

The export price of Thai and Chinese shrimp products was the leading factor that affected the export price for Vietnam, and this factor had the top ranking in the SHAP evaluation. Thailand is famous for the quality of its shrimp products. Seafood is the industry that generates most income for Thailand, and frozen shrimp are the highest value export from the country. The FAO [54] reported that Thailand was the top global exporter of shrimp products, which were the county's most important commodity trade in terms of value. About 82% of the shrimp produced is used for export, while the remaining 18% is consumed domestically. In the production of shrimp, Thailand has prioritized food safety, welfare, and traceability among shrimp farmers and its production is conducted in an environmentally responsible manner. The US is the most important importer of Thailand shrimp products and Thailand is a strong competitor of Vietnam in the US shrimp market. With the advantage of its large potential farming area, China could also produce a large quantity of shrimp and it has become the world's largest producer of shrimp [2]. The Chinese shrimp export price was the lowest among the exporters investigated here (as shown in Fig 5), which had an impact on the export price in Vietnam. Chile is also a strong competitor with Vietnam in exporting shrimp to the US market

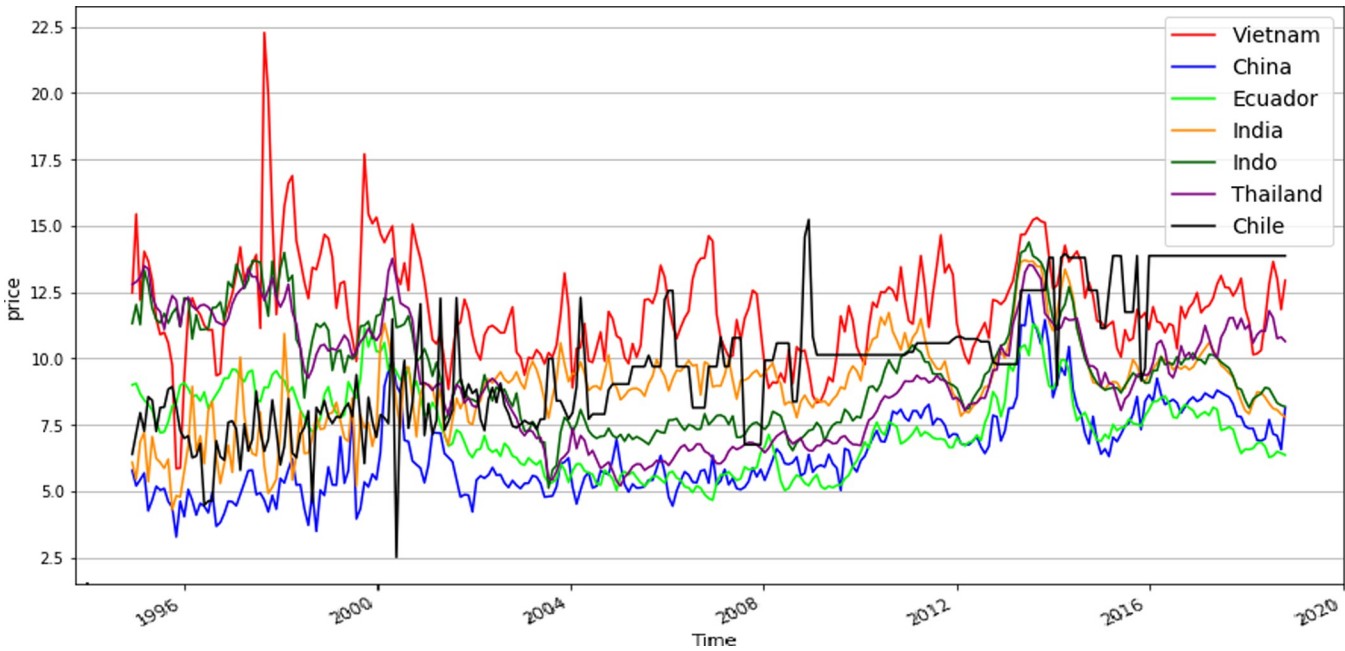

**Fig 5. Comparison of prices among competitors, including Vietnam.**

due to its advantageous geographical location. Chile and the US are located in the same continent, which reduces transportation costs and there are fewer trade barriers imposed by the US, i.e., the anti-dumping law. Chile also has favorable natural conditions, i.e. a long coastline, which favors shrimp production.

As a member of the WTO, Vietnam has obtained an advantage in exporting shrimp to the US market, which has partly overcome the unfair imposition of the anti-dumping tariff from the US government that is used to protect inefficient domestic industries [7]. Vietnam became the 150[th] member of WTO in 2007, and the export volume and price of shrimp products was positively affected. It increased from 11.78 USD/kg (Dec-2006) to 13.00 USD/kg (Jan-2007). In addition, WTO members face lower trade barriers; thus, they obtain more benefits from low tariffs, regulations, and import quotas.

Disease clearly influenced the export price. It not only affected the quality of shrimp but also caused a scarcity in the quantity for export. The global shrimp industry has been severely affected by early mortality syndrome, and has experienced huge losses. In addition to the decrease in production for domestic consumption, shrimp losses have directly affected national exports by causing fluctuations in the exported volume and price. According to data of the US Department of Agriculture used in this study, the average export price of Vietnamese shrimp to the US was 11.7 USD/kg, which increased to 12.2 USD/kg after early mortality syndrome was observed. Similarly to Vietnam, Thailand has also experienced large changes in the export price due to the early mortality syndrome outbreak. The average price before the disease outbreak was 8.8 USD/kg, but this increased to 10.3 USD/kg after infection. The Chinese export price increased from 5.5 to 7.8 USD/kg after EMS was confirmed, while the shrimp product from Chile increased by an average of 4.7 USD for each exported kilogram, which resulted in an overall increase from 8.3 to 13.0 USD/kg. The early mortality syndrome outbreak clearly affected the export prices of shrimp products in the global market. The disease reduced the production volume; thus, it caused the global price to increase. Another impact of the disease on global trade was the reduction of opportunities for exports. Importers could

implement policies restricting imported shrimp from affected countries [8]. This will reduce the competitiveness and export volume of exporters.

Although other factors, including Aquaculture Steward Council certificates, global GAP, Safe Quality Food, and Hazard Analysis Critical Control Point, had less of an impact on the predictive model, they also impacted the export price of Vietnam and other countries in terms of the assurance of food safety, traceability, and disease risk. It is likely that Vietnam will obtain a better price for its shrimp products if it fully implements the assurance certificates for exported shrimp, which may give it a competitive advantage over other producer countries in the international market. The more the requirements of the US are satisfied, the greater the export volume and price Vietnam will obtain. These certificates not only help increase productivity and reduce the risks from diseases during stocking, but they also ensure the safety of exported foods [55]. Currently, the export price of Vietnamese shrimp is higher than that of other exporters (as shown in Fig 5). This presents difficulties for Vietnam in terms of competition with other countries in the US market. The quality of shrimp products is a significant consideration during farming due to the enhancement of product quality being the main goal for the Vietnamese shrimp production industry. This explains why the US still prefers to import shrimp products from Vietnam even though the price is higher than products from other countries. Vietnam is one of the major exporting countries to the US market; however, the requirements imposed on Vietnamese producers by the US market have increased the cost of production. For example, Hazard Analysis Critical Control Point, a traceability certificate, is considered the best strategy for gaining consumer trust regarding exported seafood products. Accordingly, Vietnam needs to implement Hazard Analysis Critical Control Point in cultured shrimp for export. However, the impact on the production cost needs to be considered [56]. To increase confidence in the origin of product, Vietnam issued a national traceability regulation (Circular No.03/2011 BNN-PTNT) through the Vietnamese Directorate of Fisheries in March-2011. Applying certificates in farming will satisfy the US market, but the product cost will need to increase to guarantee a profit for farmers. Currently, the export price of Vietnam's shrimp is 30% higher than that of Ecuador, India, and Indonesia [57]. Therefore, planning the good strategy for suitable export price and increasing product quality will make Vietnam's shrimp products more competitive with other exporters, leading to an increase in the ranking of the country among the top exporting countries in the future.

## Supporting information

**S1 Dataset.**
(CSV)

## Author Contributions

**Conceptualization:** Nguyen Minh Khiem, Yuki Takahashi, Nobuo Kimura.

**Data curation:** Nguyen Minh Khiem, Yuki Takahashi, Khuu Thi Phuong Dong.

**Formal analysis:** Nguyen Minh Khiem, Yuki Takahashi, Nobuo Kimura.

**Investigation:** Nguyen Minh Khiem, Yuki Takahashi, Hiroki Yasuma, Khuu Thi Phuong Dong, Tran Ngoc Hai, Nobuo Kimura.

**Methodology:** Nguyen Minh Khiem, Yuki Takahashi, Nobuo Kimura.

**Resources:** Nguyen Minh Khiem, Yuki Takahashi, Hiroki Yasuma, Khuu Thi Phuong Dong.

**Software:** Nguyen Minh Khiem, Yuki Takahashi.

**Supervision:** Yuki Takahashi, Hiroki Yasuma, Tran Ngoc Hai, Nobuo Kimura.

**Writing – original draft:** Nguyen Minh Khiem, Yuki Takahashi, Khuu Thi Phuong Dong.

**Writing – review & editing:** Nguyen Minh Khiem, Yuki Takahashi, Hiroki Yasuma, Khuu Thi Phuong Dong, Tran Ngoc Hai, Nobuo Kimura.

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
