## [Decision Letter · Decision Letter 0]

20 Jun 2022

PONE-D-22-12822A novel machine learning approach to predict the export price of seafood products based on competitive information: evidence from the export of Vietnamese shrimp to the US market PLOS ONE

Dear Dr. Nguyen,

Thank you for submitting your manuscript to PLOS ONE. After careful consideration, we feel that it has merit but does not fully meet PLOS ONE’s publication criteria as it currently stands. Therefore, we invite you to submit a revised version of the manuscript that addresses the points raised during the review process.

We look forward to receiving your revised manuscript.

Kind regards,

José F. Vicent Francés, Ph.D.

Academic Editor

PLOS ONE

Journal Requirements:

"This work was supported by the Hokkaido University DX Doctoral Fellowship [grant number JPMJSP2119]."

Additional Editor Comments :

Due to reviewer feedback, my decision is Major Revision

Reviewers' comments:

Reviewer's Responses to Questions

**Comments to the Author**

1. Is the manuscript technically sound, and do the data support the conclusions?

Reviewer #1: Partly

Reviewer #2: Partly

2. Has the statistical analysis been performed appropriately and rigorously? 

Reviewer #1: No

Reviewer #2: No

3. Have the authors made all data underlying the findings in their manuscript fully available?

Reviewer #1: Yes

Reviewer #2: No

4. Is the manuscript presented in an intelligible fashion and written in standard English?

Reviewer #1: Yes

Reviewer #2: Yes

5. Review Comments to the Author

Reviewer #1: In this paper, the authors use a super learner, with the aim of to give accurate and stable price predictions for Vietnamese shrimp products exported to the US market. In the paper, 10 algorithms are combined to predict the exported price of Vietnamese shrimp based on information from competitive exporters as China, Thailand, Indonesia, or India. In addition, a SHAP method is used to determine how each variable (predictor) influenced in the price.

In my opinion, the paper is interesting and novel but very simple. There are some technical weaknesses that must be solved and it will help clarify the visibility of the paper.

The use of acronyms is abused, which makes reading difficult. Please use them in necessary cases but not continuously.

First of all, the paper would have to be reorganized:

The Materials and Method section should be divided into Preliminaries, where all the methods and dataset used in the paper are included and Methodology, where they put

I am not sure if the variables the authors use (in table 1) are correlations or simple differences since the name is CorrelationPrice but the description talks about differences. It is mandatory to clarify that.

In line 209 the authors say that “After evaluating the suitability of the combination of algorithms…” but: How do the authors assess the adequacy of these algorithms? Why choose 10 algorithms if it is known in advance that some of them will not give precise results?

The inclusion of an explanatory flow chart is mandatory.

I don't understand the title of the Section “Results Machine Learning”. It must be changed.

Figures 2, 3, 4 and 5 are practically the same. That means that the predictions at 3, 6, 9 and 12 months based on super learner are practically identical. Is this temporary specification necessary?

Regarding the data, nothing is said about the amount of data used, nothing is said about data cleaning, nothing is said about whether this data is balanced or not.

An important weakness of the paper is the technical section. Then, nothing is said about the different types of algorithms used (they only describe a simple definition as Wikipedia) . They must specify much more the technical part of the data, the technical part of the algorithms used as well as the technical part of the super-learne. They are treated very weakly. For example, authors do not provide any table or description with the training parameters and the hyper-parameters of each model. This must be explicitly indicated.

Last, since the problem is a timeseries forecasting task, why authors have not used well-known algorithms in this field such as ARIMA, VAR or VARMAX models?

Reviewer #2: The authors use a super learner (regression model) as a predictor of Vietnamese shrimp prices as well as the SHAP method to estimate the most important features/factors that influence the predicted variable. The authors report strong results in the accuracy of the predictions and the feature importance analysis allows the highlight of useful information from the trained model. Their approach is interesting since the ability to propose (i) an accurate prediction of shrimp prices and (ii) the reasons behind the model prediction (or feature importance) can both be useful for producers or policymakers to develop more adapted strategies in the future.

Comment on data availability:

The PLOS Data policy requires authors to make all data underlying the findings described in their manuscript fully available without restriction. The authors did not make all data used in their study fully available which thus prevents reviewers or readers from independently verifying the validity of their results. The authors write that the "Data cannot be shared publicly because it is sensitive and relates to Vietnam and other countries economics sector." but then that they have "we have some papers which relate this data, but have not been published yet". For me, these two statements may appear contradictory. Moreover, the data used in their study (which includes price, required farming certificates, and disease outbreaks from May-1995 to May-2019) seems to be published by international organizations such as the US Department of Agriculture (USDA), the Food and Agriculture Organization of the United Nations (FAO), and the World Trade Organization (WTO), and publicly available to consult or download. For example, the FAO statistics that are part of the data used by the authors seem to be publicly available (e.g. Global aquaculture production can be downloaded at https://www.fao.org/fishery/en/collection/aquaculture?lang=en). In my opinion, in order to show the reproducibility of their results, the authors should provide clear references to download the data as well as the source code of the scripts they used for data cleaning, features extraction, and classification. If, for reasons that should be approved by the journal, the authors are not authorized to reproduce and/or make the raw data used in their study available, they should at least provide an archive containing the extracted features used to train their models.

Comment on the reproducibility of results:

The authors did not detail enough the methodology used in their work in order to make it easily replicable. For each of their base models, the authors must also provide all information and/or parameters in order for the readers to be able to reproduce their results. A few examples include (but are not limited to) the depth of the decision tree, number of trees in the random forest (and other models), number of nodes per layer, error function, learning rate, optimization function.. for the neural network, regularization parameter C in the SVR, etc... Moreover, the authors should also specify if they were default parameters or, if not, how those parameters were selected. The source code of the scripts they used for data cleaning, features extraction and classification would greatly help in this regard.

In this state, the paper does not offer the required information (data and methodology) to replicate the proposed results and that essential information constitutes one of the most important prerequisites that should be met before this work can be accepted for publication.

Other comments about the paper:

1. The title says "evidence", but it is not clear what kind of evidence it's referring to. Moreover, the super learner approach is hardly novel in machine learning and should not be referred to as such in a general sense. If the authors referred to the first time a super learner is used to specifically predict shrimp prices, they should be more precise so that the reader is not misled to think that the super learner proposed by the authors is a novel approach in machine learning.

2. Page 8, "Finally, we obtained 13 independent variables for use in the super learner process to predict the Vietnamese export price": the analysis resulting in the choice of variables is interesting and is clearly described, however, how did the authors determine that these 13 variables were independent? The authors should specify the statistical tests that were performed, describe the methods used to select these tests, and publish the precise results of their statistical analysis that lead to the conclusion that these 13 variables were indeed independent.

3. Page 8: moreover, in addition to the analysis proposed to determine the choice of selected variables, the authors should also support their choice by performing a statistical analysis or a forward feature selection (or any other traditional feature selection technique, e.g. see https://doi.org/10.1016/B978-0-444-81892-8.50040-7). These kinds of analyses are helpful to quantify the importance of all variables and validate the choice proposed by the authors or discuss why the choice proposed by the authors varies from these more traditional methods. For example, in the discussion, the authors state that the price of Indian shrimps significantly affects the price of Vietnamese shrimps. It would interesting to perform a correlation analysis between the price of India and Vietnam and report the result. Additionally, a forward feature selection analysis (and the evolution of prediction accuracy when incrementally adding more features) will also show if the model would already perform well with a smaller number of features (or even with just the price of Indian shrimps) or if adding more features was essential to reach the excellent performances reported by the authors. This type of analysis becomes increasingly important in machine learning and, in my opinion, should be systematically performed.

4. Pages 10-14: if the target audience is machine learning experts, the description of the base models written by the authors is too general, often simplistic, and offers little useful information (no detailed explanation on the choice of parameters, model architecture, why the model was specifically chosen, etc..). If the target audience is not supposed to have prior knowledge of machine learning, then the descriptions should be easy and clear to understand. But at the risk of repeating myself, in addition to the model's description, authors are required to provide all the necessary information about their models to ensure the reproducibility of their results.

5. Page 11: Formula (1): please specify what p and n stand for. Formula (2): I would suggest writing beta under the argmin function to make it explicit that the argmin function finds the beta that minimizes the sum. Please also specify the N parameter so it doesn't get confused with the n of the first formula (you may also consider changing the n of Formula (1)).

6. As the authors correctly stated, a super learner model is at least as good, and (hopefully) better, than any individual models that compose it. Thus, comparing the super learner results to the candidate models does not give any information to the reader concerning the performance of the authors' approach compared to state-of-the-art models for price predictions. I think that it would have been much more interesting and informative, in order to evaluate the performance of their approach, if the authors would also have compared the super-learning results with state-of-the-art models for price predictions such as LSTM/GRU, ARIMA, etc.. (as done in https://doi.org/10.1007/s00521-020-05172-3).

7. Page 21: the sentence "These performances proved that the super learner is a novel approach that produced the predictions in this study." should be rephrased. Did the authors want to emphasize that the super learner produced the best predictions? Plus, although the authors demonstrated the superior results of the super learner compared to any of its composing predictors, as stated earlier, the super learner approach is hardly novel in machine learning and should not be referred to as such in a general sense by the authors.

8. Line 460: "contributing 70% of the value of India’s exports": Did the authors want to say "India's seafood exports"? Because the sentence tends to state that it's 70% of India's total exports.

9. Figure 2-5: price should have an uppercase 'p'

6. PLOS authors have the option to publish the peer review history of their article (what does this mean?). If published, this will include your full peer review and any attached files.

Reviewer #1: No

Reviewer #2: No

---

## [Author Response · Author response to Decision Letter 0]

15 Jul 2022

Reviewer #1: In this paper, the authors use a super learner, with the aim of to give accurate and stable price predictions for Vietnamese shrimp products exported to the US market. In the paper, 10 algorithms are combined to predict the exported price of Vietnamese shrimp based on information from competitive exporters as China, Thailand, Indonesia, or India. In addition, a SHAP method is used to determine how each variable (predictor) influenced in the price.

In my opinion, the paper is interesting and novel but very simple. There are some technical weaknesses that must be solved and it will help clarify the visibility of the paper.

The use of acronyms is abused, which makes reading difficult. Please use them in necessary cases but not continuously.

Answer: Thank you very much for your comment. We have reduced the number of acronyms in the manuscript.

First of all, the paper would have to be reorganized:

The Materials and Method section should be divided into Preliminaries, where all the methods and dataset used in the paper are included and Methodology, where they put

Answer: Thank you very much for your comment. We reorganized the Materials and Method according to your suggestion in the manuscript.

I am not sure if the variables the authors use (in Table 1) are correlations or simple differences since the name is CorrelationPrice but the description talks about differences. It is mandatory to clarify that.

Answer: Thank you very much for your comment. There is a difference between the price of Vietnamese shrimp and the price of other countries. But the difference is calculated in one direction, the price of Vietnam minus the price of others in each monthly dataset. Therefore, the data contain both negative and positive values. If a value is positive, it means that the Vietnam price is higher. If the value is negative, it means that the Vietnam price is lower. Both the value and direction of these differences will contribute to the prediction. Also, the correlation of distribution between Vietnamese export price and that of other countries is shown in Fig 1. 

We changed the name “CorrelationPrice” to “DifferencePrice” in Table 1 to match with the description. 

In line 209 the authors say that “After evaluating the suitability of the combination of algorithms…” but: How do the authors assess the adequacy of these algorithms? Why choose 10 algorithms if it is known in advance that some of them will not give precise results?

The inclusion of an explanatory flow chart is mandatory.

Answer: Thank you very much for your comment. In the previous paper (https://doi.org/10.1007/s12562-021-01498-6), random forest and gradient boosting were the best single machine learning approaches to predict the price of Vietnamese export shrimp. In the super learner, the combination of algorithms is required to generate an ensemble model. Therefore, the selected set of algorithms, called S, contains random forest and gradient boosting. We iteratively add new algorithms into the current set, S. To evaluate the new potential algorithm, we tried adding it in S to test the accuracy of the ensemble model. If it improved the accuracy, we added this new algorithm into S. Otherwise, we eliminated it and chose another algorithm. The more algorithms we added, the more accurate the ensemble model became. However, too many candidate algorithms will increase the time and computer cost for implementation. Here, we set the maximum number of algorithms to 10 for the current dataset. The forward selection of the algorithms is specified in Table 2. We added a description of this process to the manuscript at lines 360 - 371.

I don't understand the title of the Section “Results Machine Learning”. It must be changed.

Answer: Thank you very much for your comment. The results section has many subsections. We modified the heading you mention to “prediction accuracy.”

Figures 2, 3, 4 and 5 are practically the same. That means that the predictions at 3, 6, 9 and 12 months based on super learner are practically identical. Is this temporary specification necessary?

Answer: Thank you very much for your comment. The prediction by the super learner has stable accuracy for different periods of data (3, 6, 9, or 12 months). Therefore, there are small differences (small errors) among periods. Figures 2, 3, 4, and 5 intend to emphasize that the use of the super learner can overcome the dependence of the algorithm on long- or short-term periods of data. To more easily compare these figures, we combined them into one figure, now Fig 3. We explain this point in the manuscript at lines 520–524.

Regarding the data, nothing is said about the amount of data used, nothing is said about data cleaning, nothing is said about whether this data is balanced or not.

Answer: Thank you very much for your comment. The monthly datasets were collected from the US Department of Agriculture, WTO, and FAO for the period from May 1995 to May 2019. Therefore, there are 289 rows of data. We only focus on the competition among 7 countries that export to the US market, including China, Thailand, India, Indonesia, Ecuador, Chile, and Vietnam. Therefore, we chose 13 variables related to competition among countries as explained at lines 137 to 149. The dataset used in this research has 289 rows and 13 columns. For machine learning, we separated 2 periods: May 1995 to Apr 2013 for the training model, and May 2013 to May 2019 for testing accuracy as described at lines 376 to 380. We also added the distribution of export prices of Vietnamese shrimp and the correlation between the Vietnamese export price and that of other countries to Fig 1. Also, an explanation of Fig 1 was added to the manuscript at lines 139 – 149. 

An important weakness of the paper is the technical section. Then, nothing is said about the different types of algorithms used (they only describe a simple definition as Wikipedia). They must specify much more the technical part of the data, the technical part of the algorithms used as well as the technical part of the super-learner. They are treated very weakly. For example, authors do not provide any table or description with the training parameters and the hyper-parameters of each model. This must be explicitly indicated.

Answer: Thank you very much for your comment. We added the description of parameters used in each algorithm in the manuscript. 

Linear regression: line 206 – 210

Lasso regression: line 224 – 228

Ridge regression: line 238 – 242

Elastic net: line 252 – 255

K-nearest neighbor: line 263 – 267 

SVR: line 278 – 282 

Decision Tree: line 291 – 294

Random forest: line 301 – 306 

Gradient Boosting: 313 – 317

Neural network: line 329 – 334 

Extra tree: line 344 – 348 

Last, since the problem is a timeseries forecasting task, why authors have not used well-known algorithms in this field such as ARIMA, VAR or VARMAX models?

Answer: Thank you very much for your comment. We did not use these algorithms (ARIMA, VAR, and VARMAX) in this paper as we already evaluated them in a previous paper (https://doi.org/10.1007/s12562-021-01498-6). Although these algorithms will give state-of-the-art models, these time series analyses were not suitable for our dataset because of large error prediction. Also, we tested VARMAX for our dataset and compared its accuracy to the current candidate algorithms. Here, we applied VARMAX under the Vector Autoregressive Moving Average with eXogenous regressors model (statsmodels.tsa.statespace.varmax.VARMAX(train_feature,order = (2,0))) using Python version 3.7. The mean absolute error for the 3-month base of data was 8.82% for the testing subset, 15.15% (for the 6-month base), 17.30% (for the 9-month base), and 21.00% (for the 12-month base), which had a larger average error, compared to linear regression. In fact, linear regression was one of worst candidate algorithms for the super learner. When we combined VARMAX with the current 10 algorithms, the error (MAPE) for the 3-month based prediction was 1.17% larger than the current error (0.80%). Therefore, these time series analyses were not added to the super learner. 

Reviewer #2: The authors use a super learner (regression model) as a predictor of Vietnamese shrimp prices as well as the SHAP method to estimate the most important features/factors that influence the predicted variable. The authors report strong results in the accuracy of the predictions and the feature importance analysis allows the highlight of useful information from the trained model. Their approach is interesting since the ability to propose (i) an accurate prediction of shrimp prices and (ii) the reasons behind the model prediction (or feature importance) can both be useful for producers or policymakers to develop more adapted strategies in the future.

Comment on data availability:

The PLOS Data policy requires authors to make all data underlying the findings described in their manuscript fully available without restriction. The authors did not make all data used in their study fully available which thus prevents reviewers or readers from independently verifying the validity of their results. The authors write that the "Data cannot be shared publicly because it is sensitive and relates to Vietnam and other countries economics sector." but then that they have "we have some papers which relate this data, but have not been published yet". For me, these two statements may appear contradictory. Moreover, the data used in their study (which includes price, required farming certificates, and disease outbreaks from May-1995 to May-2019) seems to be published by international organizations such as the US Department of Agriculture (USDA), the Food and Agriculture Organization of the United Nations (FAO), and the World Trade Organization (WTO), and publicly available to consult or download. For example, the FAO statistics that are part of the data used by the authors seem to be publicly available (e.g. Global aquaculture production can be downloaded at https://www.fao.org/fishery/en/collection/aquaculture?lang=en). In my opinion, in order to show the reproducibility of their results, the authors should provide clear references to download the data as well as the source code of the scripts they used for data cleaning, features extraction, and classification. If, for reasons that should be approved by the journal, the authors are not authorized to reproduce and/or make the raw data used in their study available, they should at least provide an archive containing the extracted features used to train their models.

Answer: Thank you for your comment. Other papers now partly use this dataset and are in the process of publication. We willing to share the original dataset to the Plos one journal that was analyzed to train the model of the super learner in this paper.

Comment on the reproducibility of results:

The authors did not detail enough the methodology used in their work in order to make it easily replicable. For each of their base models, the authors must also provide all information and/or parameters in order for the readers to be able to reproduce their results. A few examples include (but are not limited to) the depth of the decision tree, number of trees in the random forest (and other models), number of nodes per layer, error function, learning rate, optimization function.. for the neural network, regularization parameter C in the SVR, etc... Moreover, the authors should also specify if they were default parameters or, if not, how those parameters were selected. The source code of the scripts they used for data cleaning, features extraction and classification would greatly help in this regard.

In this state, the paper does not offer the required information (data and methodology) to replicate the proposed results and that essential information constitutes one of the most important prerequisites that should be met before this work can be accepted for publication.

Answer: Thank you very much for your comment. We added the description of parameters for each algorithm. 

Linear regression: line 206 – 210

Lasso regression: line 224 – 228

Ridge regression: line 238 – 242

Elastic net: line 252 – 255

K-nearest neighbor: line 263 – 267 

SVR: line 278 – 282 

Decision Tree: line 291 – 294

Random forest: line 301 – 306 

Gradient Boosting: 313 – 317

Neural network: line 329 – 334 

Extra tree: line 344 – 348 

Other comments about the paper:

1. The title says "evidence", but it is not clear what kind of evidence it's referring to. Moreover, the super learner approach is hardly novel in machine learning and should not be referred to as such in a general sense. If the authors referred to the first time a super learner is used to specifically predict shrimp prices, they should be more precise so that the reader is not misled to think that the super learner proposed by the authors is a novel approach in machine learning.

Answer: Thank you very much for your comment. The word “evidence” may cause confusion for the reader. Here, the “evidence” in the title implies the case of Vietnamese shrimp export. It does not mean the evidence for the super learner. The super learner was highly accurate when it was applied to predict the dataset of Vietnamese exports. We changed “evidence from” to “the case of” in the title. 

2. Page 8, "Finally, we obtained 13 independent variables for use in the super learner process to predict the Vietnamese export price": the analysis resulting in the choice of variables is interesting and is clearly described, however, how did the authors determine that these 13 variables were independent? The authors should specify the statistical tests that were performed, describe the methods used to select these tests, and publish the precise results of their statistical analysis that lead to the conclusion that these 13 variables were indeed independent.

Answer: Thank you very much for your comment. To determine the independent variable used for hypothesizing the export price of Vietnamese shrimp, we applied the Pearson correlation method. In machine learning, the Pearson correlation method is popularly used to select variables/features for prediction (https://doi.org/10.1260/1748-3018.6.3.385). We calculated the association between influence variables and the Vietnamese export price. The Pearson correlation assigns a value between −1 and 1, where 0 is no correlation, 1 is total positive correlation, and −1 is total negative correlation. Therefore, an absolute value of correlation from 0.5 to 1 is mostly acceptable and considered a high association with the target value. Although correlation between price Thailand and Vietnam is 0.47 (<0.5), we also used it to fully evaluate the competition. These 13 variables satisfy the criteria of the Pearson correlation. Therefore, they were used in the prediction. Also, we explained the method of selecting the variables in the manuscript at lines 188 – 197 and the results of the selection at lines 420–431.

3. Page 8: moreover, in addition to the analysis proposed to determine the choice of selected variables, the authors should also support their choice by performing a statistical analysis or a forward feature selection (or any other traditional feature selection technique, e.g. see https://doi.org/10.1016/B978-0-444-81892-8.50040-7). These kinds of analyses are helpful to quantify the importance of all variables and validate the choice proposed by the authors or discuss why the choice proposed by the authors varies from these more traditional methods. For example, in the discussion, the authors state that the price of Indian shrimps significantly affects the price of Vietnamese shrimps. It would interesting to perform a correlation analysis between the price of India and Vietnam and report the result. Additionally, a forward feature selection analysis (and the evolution of prediction accuracy when incrementally adding more features) will also show if the model would already perform well with a smaller number of features (or even with just the price of Indian shrimps) or if adding more features was essential to reach the excellent performances reported by the authors. This type of analysis becomes increasingly important in machine learning and, in my opinion, should be systematically performed.

Answer: Thank you very much for your comment. We chose the Pearson correlation method to select variables with a strong influence on the target value–Vietnamese export price. Due to the absolute value of correlation ranges from 0 to 1, a value of correlation from 0.5 to 1 is acceptable and considered to represent a high impact on the target value. Most variables satisfied this condition. The price of Indian shrimp strongly influenced the Vietnam price, showing a correlation value of 0.89, the highest among all countries (Fig 1). Also, the distribution of the Indian price is more similar to the Vietnam price but lower. The Indian price was mainly 8–11 USD, while the Vietnam price was 10–15 USD. This means that India will have a competitive advantage over the Vietnam price, which was estimated in the SHAP evaluation (Fig 4). 

The explanation regarding the competition of Indian prices in the Discussion is at lines 540 – 549. 

4. Pages 10-14: if the target audience is machine learning experts, the description of the base models written by the authors is too general, often simplistic, and offers little useful information (no detailed explanation on the choice of parameters, model architecture, why the model was specifically chosen, etc..). If the target audience is not supposed to have prior knowledge of machine learning, then the descriptions should be easy and clear to understand. But at the risk of repeating myself, in addition to the model's description, authors are required to provide all the necessary information about their models to ensure the reproducibility of their results.

Answer: Thank you very much for your comment. We described the parameters of each algorithm in the manuscript. We also explain how to select these algorithms at lines 360–371.

5. Page 11: Formula (1): please specify what p and n stand for. Formula (2): I would suggest writing beta under the argmin function to make it explicit that the argmin function finds the beta that minimizes the sum. Please also specify the N parameter so it doesn't get confused with the n of the first formula (you may also consider changing the n of Formula (1)).

Answer: Thank you very much for your comment. We modified equations (1) and (2) in the manuscript. Now, they become equations (2) and (3), respectively.

6. As the authors correctly stated, a super learner model is at least as good, and (hopefully) better, than any individual models that compose it. Thus, comparing the super learner results to the candidate models does not give any information to the reader concerning the performance of the authors' approach compared to state-of-the-art models for price predictions. I think that it would have been much more interesting and informative, in order to evaluate the performance of their approach, if the authors would also have compared the super-learning results with state-of-the-art models for price predictions such as LSTM/GRU, ARIMA, etc… (as done in https://doi.org/10.1007/s00521-020-05172-3).

Answer: Thank you very much for your comment. As shown in our previous paper (https://doi.org/10.1007/s12562-021-01498-6), we already tried time series forecasting algorithms to find the best single algorithm for prediction, but we did not select them. Although these algorithms are state-of-the-art models, they seem to be unsuitable for our dataset due the larger error compared to machine learning algorithms. In this study, we again tested the VARMAX algorithm using the Vector Autoregressive Moving Average with eXogenous regressors model (statsmodels.tsa.statespace.varmax.VARMAX(train_feature,order = (2,0))) with Python version 3.7. The mean absolute error for the 3-month base of data was 8.82% for the testing subset, 15.15% (for the 6-month base), 17.30% (for the 9-month base), and 21.00% (for the 12-month base), which had a larger average error, compared to linear regression. In the super learner, the compatibility among algorithms is important. We also combined VARMAX with the current 10 algorithms. The error (MAPE) is 1.17% for the 3-month base period (larger than the current 10 algorithms which was 0.80%). Therefore, these time series forecasting algorithms were not added in the super learner. 

7. Page 21: the sentence "These performances proved that the super learner is a novel approach that produced the predictions in this study." should be rephrased. Did the authors want to emphasize that the super learner produced the best predictions? Plus, although the authors demonstrated the superior results of the super learner compared to any of its composing predictors, as stated earlier, the super learner approach is hardly novel in machine learning and should not be referred to as such in a general sense by the authors.

Answer: Thank you very much for your comment. We rephrased this sentence in manuscript at lines 521–524.

8. Line 460: "contributing 70% of the value of India’s exports": Did the authors want to say "India's seafood exports"? Because the sentence tends to state that it's 70% of India's total exports.

Answer: Thank you very much for your comment. We modified India’s exports to India’s seafood exports (line 547 – 548).

9. Figure 2-5: price should have an uppercase 'p'

Answer: Thank you very much for your comment. We changed ‘price’ to ‘Price’ in Fig 3 (before was Fig 2–Fig 5)

---

## [Decision Letter · Decision Letter 1]

13 Sep 2022

A novel machine learning approach to predict the export price of seafood products based on competitive information: The case of the export of Vietnamese shrimp to the US market

PONE-D-22-12822R1

Dear Dr. Nguyen,

We’re pleased to inform you that your manuscript has been judged scientifically suitable for publication and will be formally accepted for publication once it meets all outstanding technical requirements.

Kind regards,

José F. Vicent Francés, Ph.D.

Academic Editor

PLOS ONE

Additional Editor Comments (optional):

Reviewers' comments:

Reviewer's Responses to Questions

**Comments to the Author**

1. If the authors have adequately addressed your comments raised in a previous round of review and you feel that this manuscript is now acceptable for publication, you may indicate that here to bypass the “Comments to the Author” section, enter your conflict of interest statement in the “Confidential to Editor” section, and submit your "Accept" recommendation.

Reviewer #2: All comments have been addressed

2. Is the manuscript technically sound, and do the data support the conclusions?

Reviewer #2: Yes

3. Has the statistical analysis been performed appropriately and rigorously? 

Reviewer #2: Yes

4. Have the authors made all data underlying the findings in their manuscript fully available?

Reviewer #2: Yes

5. Is the manuscript presented in an intelligible fashion and written in standard English?

Reviewer #2: Yes

6. Review Comments to the Author

Reviewer #2: I believe the authors have adequately addressed the comments raised in the previous round of review. I particularly appreciate that the authors made their dataset fully available and provided all technical information to ensure the reproducibility of their results. For me, this manuscript is now acceptable for publication. Thank you to the authors for their interesting work.

7. PLOS authors have the option to publish the peer review history of their article (what does this mean?). If published, this will include your full peer review and any attached files.

Reviewer #2: **Yes: **Cédric Simar

---

## [Editor Report · Acceptance letter]

19 Sep 2022

PONE-D-22-12822R1 

A novel machine learning approach to predict the export price of seafood products based on competitive information: The case of the export of Vietnamese shrimp to the US market 

Dear Dr. Nguyen:

I'm pleased to inform you that your manuscript has been deemed suitable for publication in PLOS ONE. Congratulations! Your manuscript is now with our production department. 

Kind regards, 

on behalf of

Dr. José F. Vicent Francés 

Academic Editor

PLOS ONE